# The RhoGEF Trio: A Protein with a Wide Range of Functions in the Vascular Endothelium

**DOI:** 10.3390/ijms221810168

**Published:** 2021-09-21

**Authors:** Lanette Kempers, Amber J. M. Driessen, Jos van Rijssel, Martijn A. Nolte, Jaap D. van Buul

**Affiliations:** 1Laboratory for Vascular Cell Biology, Department of Molecular Hematology, Sanquin Research, Plesmanlaan 125, 1066 CX Amsterdam, The Netherlands; l.kempers@sanquin.nl (L.K.); amberdriessen1997@gmail.com (A.J.M.D.); j.vanrijssel@sanquin.nl (J.v.R.); 2Landsteiner Laboratory, Amsterdam UMC, University of Amsterdam, 1066 CX Amsterdam, The Netherlands; 3Research Facility, Sanquin Research and Landsteiner Laboratory, 1066 CX Amsterdam, The Netherlands; m.nolte@sanquin.nl; 4Leeuwenhoek Centre for Advanced Microscopy, Section of Molecular Cytology, Swammerdam Institute for Life Sciences, University of Amsterdam, 1066 CX Amsterdam, The Netherlands

**Keywords:** Rho-GEF, small GTPase, vasculature, endothelium, inflammation

## Abstract

Many cellular processes are controlled by small GTPases, which can be activated by guanine nucleotide exchange factors (GEFs). The RhoGEF Trio contains two GEF domains that differentially activate the small GTPases such as Rac1/RhoG and RhoA. These small RhoGTPases are mainly involved in the remodeling of the actin cytoskeleton. In the endothelium, they regulate junctional stabilization and play a crucial role in angiogenesis and endothelial barrier integrity. Multiple extracellular signals originating from different vascular processes can influence the activity of Trio and thereby the regulation of the forementioned small GTPases and actin cytoskeleton. This review elucidates how various signals regulate Trio in a distinct manner, resulting in different functional outcomes that are crucial for endothelial cell function in response to inflammation.

## 1. Introduction

Endothelial cells line the inside of blood vessels and form a semi-permeable barrier that regulates the transport of molecules and passage of immune cells to the tissues [1]. Regulation of endothelial permeability is achieved through cell–cell junctions, which are maintained by actin cytoskeletal organization and adhesion molecules. Barrier integrity can be mediated by many different stimuli, ranging from chemical and biological stimuli to mechanical stress [2]. The key regulators of the actin cytoskeleton and, thus, of endothelial permeability are small GTPases from the Rho family [3]. Rho GTPases are proteins that are involved in many cellular processes, among which include cell motility and adhesion. They oscillate between an inactive GDP-bound and active GTP-bound state. The exchange of GDP for GTP is catalyzed by Rho guanine nucleotide exchange factors (RhoGEFs), while GTPase activating proteins (RhoGAPs) perform the opposite, i.e., hydrolyzing GTP to GDP and thereby turning the protein into an inactive state [4]. The RhoGEF family contains around 70 members [5]. A RhoGEF that has shown its importance in the endothelial barrier function is Trio. Trio is part of the Dbl family of RhoGEFs, characterized by a Dbl-homologous (DH) and a pleckstrin-homology (PH) domain in their enzymatic GEF domains [5]. A unique feature of Trio is that it has three enzymatic domains (hence its name Trio). It contains two separate GEF domains, GEF1 and GEF2, that interact with and activate the small GTPases Rac1/RhoG and RhoA, respectively. The third enzymatic domain of Trio is a serine/threonine kinase domain, which specifically phosphorylates serine/threonine amino acid residues [5,6,7,8]. In addition, Trio contains a Sec14 domain, several spectrin repeats and 2 SRC Homology 3 Domains (SH3) [9,10,11,12]. Based on the amino acid build-up, we suspect Trio contains nine spectrin repeats, in contrast with other literature. Further proof is necessary to confirm or deny this claim. Isoforms of Trio can be formed due to alternative splicing (Figure 1). These isoforms, TrioA-E, have been detected in different parts of the brain and consist of either one or both of the GEF domains [13,14,15]. Based on the wide expression of Trio full length, other tissues are likely to express Trio isoforms as well, although this has not been reported so far. Our preliminary data show that at least TrioB is expressed in endothelial cells, which is confirmed via PCR (unpublished data). Differences in Trio isoform expression might play a role in the regulation of Rac1 and RhoA activity [14]. In addition, full length Trio is ubiquitously expressed in the endothelium of all organs [16]. Compared to VE-cadherin (an endothelial cell marker), Trio expression is relatively low in all organs. In recent years, the function of Trio has appeared essential for the regulation of endothelial cell morphology and endothelial barrier function. In this review, we focus on the function of Trio signaling in the healthy endothelium and how Trio may be deregulated in vascular-related inflammatory diseases.

## 2. Trio Localization

How RhoGEFs and GAPs are regulated and activated remains largely unknown. Many studies focus on a single GTPase and GEF/GAP combination. However, there are 145 RhoGEFs and GAPs currently known that regulate 10 classical Rho family GTPases [17]. This at least suggests that there must be some sort of specific regulation. Muller and coworkers studied the regulation of the three GTPases Rac1, RhoA and CDC42 and focused on the localization of the different GEFs and GAPs. By using overexpression studies and fluorescent microscopy, including FRET-based GTPase sensors, they show that RhoA-associated GEFs mainly reside near actin structures, while Rac1 GEFs are found at focal adhesion regions [18]. These findings suggest that dedicated GEFs show a close proximity to their target in order to exert their function. This is in accordance with the literature that states that Rho signaling responses are highly localized and are observed in distinct cellular zones [19,20,21,22,23,24]. Discrepancies exist with respect to the precise cellular localization of Trio. In the literature, it is has been described to localize at the plasma membrane [25,26,27], cell–cell junctions [28,29], the cytosol and vesicles [30,31] and focal adhesions or actin filaments [18]. It should be mentioned that all these studies used different cells to investigate Trio localization. Our group have reported the expression and localization of Trio in Human Umbilical Vein Endothelial cells (HUVEC) [17,20,21]. The membrane-associated localizations are corroborated by the fact that Trio contains a lipid-binding SEC14 domain. SEC14 domains have been associated with intracellular trafficking [30,32]. In mice, the isoform of Trio lacking the second GEF domain, called solo/Trio8, has shown to colocalize with early endosomes in the brain. Depletion of this isoform disrupts early endosome formation and neurite outgrowth [30]. Similarly, the SEC14 domain in p50RhoGAP, one of the first GAPs discovered, was shown to be responsible for the localization of this GAP to early endosomes [32]. Whether this is also the case for the SEC14 domain of Trio in endothelial cells remains unknown.

## 3. Trio Remodels the Actin Cytoskeleton

The Rho GTPases that are directly activated by Trio are Rac1, RhoG and RhoA, [6,7]. These Rho GTPases regulate the remodeling of the actin cytoskeleton, thereby changing cell morphology and migration. The enzymatic activity of the N-terminal TrioGEF1 induces Jun Kinase (JNK) activation through the mitogen activated protein kinase (MAPK) pathway, which are both downstream targets of Rac1 [7]. This results in the formation of membrane ruffles and lamellipodia, a key characteristic of migrating cells (Figure 2) [7]. These ruffles are not only induced by Rac1, but RhoG activation via Trio can also induce such ruffles. In the case of RhoG, these ruffles appear dorsally. It has been shown that in rat smooth muscle cells, RhoG can induce these dorsal ruffles independently of Rac1 [33]. RhoG has a higher binding affinity for Trio compared to Rac1 due to more effective GDP/GTP switching [8,34]. Whether Trio activates both GTPases individually or sequentially remains unclear. Interestingly, it has been reported that RhoG activation was necessary for the Rac1 activation during fibroblast spreading [35]. For this signaling axis, it was shown that RhoG directly interacted with the Dock180-binding protein Elmo. We showed that Trio triggered cell spreading on fibronectin by directly activating Rac1 in the absence of RhoG [36]. In the same study, we showed that downstream from Trio, RhoG could also mediate cell spreading in the absence of Rac1. In addition, in endothelial cells and under inflammatory conditions, ICAM-1 clustering triggers both Rac1 and RhoG in a Trio-dependent manner [37]. From these experiments, it can be concluded that the activation of either RhoG or Rac1 may depend on the signal induced by extracellular environment such as extracellular matrix proteins (fibronectin) and can differ between cell types. Thus far, the precise mechanisms of differential activation of Rac1 and RhoG by Trio are unclear.

In addition to RhoG and Rac1, RhoA can be activated by the GEF2 domain of Trio. This activation contributes to the formation of actin stress fibers (Figure 2) [7]. Although both the activations of Rac1/RhoG and RhoA are involved in cell migration and morphology, increased RhoA expression is linked to cell contraction, while Rac1 and RhoG are important in cell spreading. These contradicting functions are tightly balanced and regulated in the cell, by are still largely unknown mechanisms. Recently, it has been shown that, during cell spreading, distinct RhoA and Rac1 activity zones exist in the leading cells’ edge [38]. In this scenario, Trio may be essential in retaining the protrusion–contraction balance during migration due to its dual function [18]. Trio-Associated Repeat on Actin (Tara), a Trio-binding protein, may potentially be involved in this regulation [27]. This protein binds F-actin as well as the N-terminal region of Trio, the latter of which was shown to inhibit the activation of Rac1 [27]. This inhibition might increase TrioGEFD2-mediated activation of RhoA, as overexpression of Tara increases the formation of stress fibers [39]. Another alternative explanation of how Trio may control the balance between Rac1 and RhoA is the presence of the PH domain in the GEF2 domain. Swapping the PH domain of GEF2 with the PH domain of the GEF1 results in an increase in the exchange on RhoA, whereas the exchange on Rac1 was reduced [40]. Under normal circumstances the GEF2 domain is inhibited by the PH domain in the GEF2 domain. Only upon specific signaling is the inhibition released, resulting in RhoA activation. Thus, PH domains within the GEF domains of Trio seem to play a crucial role in the exchange activity of the TrioGEF domains and, therefore, in the regulation of cell morphology and migration.

## 4. The GEF Domains of Trio Differentially Regulate Endothelial Barrier Function

### 4.1. Stabilization of the Barrier

As previously mentioned, Trio is important for maintaining vascular integrity by regulating the endothelial barrier function. In order to maintain this barrier, endothelial cells develop cell–cell junctions that need constant signaling to maintain their cell–cell contacts. Two main type of junctions that link endothelial cells together are the so-called adherens junctions and tight junctions [41]. During initial assembly, adherens junctions are formed first and are important for strengthening cell–cell adhesion. These adherens junctions support the formation of tight junctions, which function mainly to regulate paracellular transport of small molecules and solutes [42]. A key step in the formation of adherens junctions is the homophilic ligation of the vascular endothelial adhesion molecule VE-cadherin. VE-cadherin is a member of the Cadherin protein family and is bound to catenins through its intracellular domain. These catenins are believed to mediate the function of cadherins [43]. Trio has shown to interact with multiple cadherins and catenins; these interactions can either stabilize or destabilize the junction strength [25,26,44]. Our group has shown that, upon ligation of VE-cadherin in vitro, Trio is recruited to VE-cadherin. This was shown by biochemical precipitation studies. The recruitment of Trio to VE-cadherin initiates the local activation of Rac1, subsequently resulting in stabilization of adherens junctions. In line with this, silencing of Trio impaired the ability of the endothelial barrier to recover in response to thrombin, most likely due to the inability to locally activate Rac1. These studies implicate an important role for Trio in not only maintenance of the adherens cell–cell junctions but also for the restoration of the endothelial barrier upon events such as inflammation [25]. Another study showed that the activation of N-cadherin in endothelial cells induces the formation of a protein complex with Trio at the plasma membrane. In this case, both Rac1 and RhoA were activated by this complex. The activation of Rac1 resulted in the recruitment of VE-cadherin to the junctions. RhoA activation further stimulated the recruitment of VE-cadherin by increasing intracellular tension, thereby reinforcing Rac1 activation (Figure 2) [45]. It is well established that pericyte coverage enhances endothelial barrier function (Figure 2) [46]. Pericytes link to endothelial cells through N-cadherin. A mouse model with an endothelial-specific knockout of N-cadherin showed increased basal vascular permeability in the lung and brain vessels, which was most likely due to a reduced density of VE-cadherin at junction regions [45]. An organ in which a correct barrier function is especially important for is the brain. Brain endothelial cells form much tighter junctions compared to other organs. This tight barrier is known to be Rac1 dependent [4,47]. Our lab previously showed that overexpression with TrioN (containing the N-terminus of Trio with only the first GEF domain) increased resistance in HUVEC via Rac1 activation [25]. This could hint towards involvement of Trio these stable junctions, but evidence is still lacking.

In physiological conditions, endothelial cells are constantly exposed to laminar shear stress due to blood flow. This induces endothelial cell alignment in the direction of flow [48]. Under flow, Trio was found to be immobilized at the cell–cell junctions and to regulate the distribution of Rac1. However, the activity of the GEFD1 domain on the N-terminus of Trio is not required for EC alignment [29]. More recently, Polacheck and colleagues showed that shear stress-dependent Dll4 signaling activates Notch1, a transmembrane receptor, in endothelial cells in order to improve the barrier function of the vasculature by catalyzing the formation of a complex containing VE-cadherin, leukocyte common antigen-related (LAR) proteins and Trio. This complex induced the local activation of Rac1, thereby promoting the assembly of adherens junctions and stabilization of the endothelial barrier [49]. An additional study showed that shear stress triggers the activation of Notch1 to signal through the cAMP/PKA pathway. This signaling axis has been found to inhibit thrombin-induced RhoA activation, making the cells less sensitive to disrupting signals [6,40].

Taking these considerations together, we can conclude that endothelial Trio is required for junction stabilization through the local activation of Rac1; therefore, it is essential for proper vascular barrier integrity.

### 4.2. Destabilization of the Endothelial Barrier by Trio

Trio signaling is not solely involved in stabilization of the barrier but can also destabilize the endothelial barrier [50]. This destabilization is established by activation of RhoA through the C-terminal GEFD2 domain of Trio, which is known for its involvement in the detachment of cells. Interestingly, the C-terminal of Trio displays autoinhibitory properties. The PH domain interacts with the DH domain at the C-terminus through hydrogen bonds, hindering the binding site for RhoA [51]. This prevents RhoA from binding to Trio and, thus, inhibits the GDP/GTP exchange on RhoA. Bandekar and colleagues revealed that the binding of Trio to the G protein coupled receptor (GPCR) Gα_q/11_ relieves this inhibition, which subsequently activates the Trio-RhoA pathway (Figure 2) [52]. Gα_q/11_ has also been shown to sense shear stress and respond to vascular leakage inducers such as thrombin, platelet activating factor and histamine [53]. Histamine itself induces Trio-mediated RhoA activation downstream of Gα_q/11_, likely by alleviating the autoinhibition of the C-terminus. This eventually resulted in focal adhesion formation, which is an adhesion site of endothelial cells to the extracellular matrix, resulting in an increase in endothelial permeability [50]. Another Gα_q/11_ signaling molecule, the sphingolipid sphingosine-1-phosphate (S1P)3, can also activate RhoA and induce stress fiber formation. This has been implied to signal transduction via Trio, but further research is needed to confirm this hypothesis [54]. S1P3 signaling also stimulates the expression of adhesion molecules, a function that could possibly signal through Trio [55]. Moreover, S1PRs have already been shown to affect endothelial permeability via activation of small GTPases through RhoGEFs [56]. In epithelial cells, Trio was found to mediate barrier destabilization through Rac1-mediated repression of E-cadherin expression. However, the protein Tara can inhibit this process by binding to the N-terminal region of Trio. As expected, a knockdown of Tara decreases the expression of E-cadherin. Nevertheless, this does not decrease the integrity of the epithelial cell sheet, most likely due to the upregulation of cadherin-6 [27].

Even though Trio has shown to be essential in the regulation of the endothelial barrier, single cell RNA sequencing by Kalucka and coworkers showed that the basal expression of Trio mRNA in mouse endothelium is relatively low [16]. Nevertheless, compared to other RhoGEFs, Trio was shown to be one of the most abundant expressed RhoGEFs in human microvascular cells [57], suggesting that low expression may be a general feature of RhoGEFs. This may indicate that enzymatic Trio activity is sufficient for maintaining the endothelial barrier. Such is the case in axon outgrowth, where low levels of Trio protein are adequate but crucial for the growth of axons [58]. Furthermore, the expression of Trio reduces when a confluent monolayer is formed in vitro [25]. This may indicate that Trio is not so much required for the maintenance of the endothelial barrier, once adherens junctions reach stability, but that they are rather more important for the initial formation of cell–cell contacts. Therefore, investigating the expression and function of endothelial Trio in inflammatory conditions may result in new insights into how endothelial Trio controls vascular homeostasis.

## 5. Sprouting Angiogenesis and Vascular Remoddeling Require the GEF1 Domain of Trio

Angiogenesis is the formation of new blood vessels from pre-existing ones, which can be divided in sprouting and non-sprouting angiogenesis [59]. An initial step in sprouting angiogenesis is the migration of so-called tip cells towards angiogenic factors, followed by stalk cells. An in vivo knockout of Trio results in embryonically lethal mice due to deformation of the skeletal muscles and neuronal disorders [60]. In this study, however, the blood vessels were not investigated, so it is unknown if these were abnormal. Nevertheless, when looking at the endothelial-specific knockout of Rac1, the mouse embryos exhibit a lack of small branched vessels [61]. Moreover, recently we showed that Trio depletion in zebrafish models drastically reduced angiogenic sprouting in developing zebrafish embryos [28]. Together, these results imply an essential and, thus, important role for Trio in the formation of new vessels through activation of the RhoGTPase Rac1, although molecular details still need to be examined. The detachments and reattachments of endothelial cells to the extracellular matrix such as collagen and laminin are crucial in the formation of new vessels. This process requires the junctions to be stable enough to maintain barrier function, while the tip cell can initiate sprouting [62,63]. Moreover, although Trio is involved in both stabilizing and destabilizing aspects of cell–cell junctions, the molecular details of how Trio may functionally regulate junctional stability during angiogenesis has thus far been unexplored and requires future studies.

Enhanced tissue metabolic requirements can result in increased vessel diameter enabled by endothelial cell enlargement. This happens, for instance, in response to an occlusion of the arteries, inducing outward remodeling of the arterial wall. Hypoxic conditions or an increase in flow can initiate this process. An in vivo zebrafish model has shown the importance of Trio signaling on arterial remodeling and endothelial cell size. Vascular endothelial growth factor (Vegf) receptor 2 signaling induced TrioGEF-1-mediated activation of Rac1 and RhoG. The expression of functionally active Trio at junctional regions facilitated local Rac1 activity, followed by the remodeling of the cytoskeleton through F-actin, which enlarged the diameter of the arterial lumen by increasing endothelial cell size but did not increase cell proliferation [28]. All in all, these studies not only show the complexity of Trio regulation but also indicate that Trio may be an important factor in arterial remodeling in ischemic diseases by increasing the size of endothelial cells.

## 6. Trio Mediates Leukocyte Adhesion for Transendothelial Migration during Inflammation

Leukocyte transendothelial migration is the process where leukocytes leave the circulation by crossing the vascular wall, i.e., the endothelial barrier and migrate into the underlying tissues. This process is triggered upon inflammatory signals and is recognized as a multistep process. It requires the combined effort of both leukocytes and endothelial cells. A critical step in this process is the expression of adhesion molecules by endothelial cells. This can be induced by inflammatory stimuli, such as LPS, TNFα and IFNγ. Aside from adhesion molecules, Trio expression also increases upon TNFα stimulation in HUVEC [55]. Our group has demonstrated that silencing Trio in vitro hampers the upregulation of the adhesion molecules VCAM-1, ICAM-1 and E-selectin upon stimulation with TNFα on both mRNA and protein levels [55]. Furthermore, we showed that the GEF1 domain of Trio was responsible for the increase in VCAM-1 expression by the direct activation of Rac1 and subsequently inducing phosphorylation and nuclear translocation of the transcription factor Ets-2 (Figure 2) [55].

For the leukocyte transmigration cascade, the following is the case: Upon leukocyte rolling and binding to the upregulated adhesion molecules on the endothelium, multiple signaling pathways in the endothelium are induced. Interactions of the leukocyte with the endothelium via ICAM-1 induces a cluster with F-actin and other actin adapter proteins [64], together triggering the appearance of a ring-like structure around the leukocyte [65]. This so-called docking structure, also termed a transmigratory cup, requires Trio, which locally mediates the activation of Rac1 and RhoG. Depletion of Trio causes malformed docking structures and a reduction in transendothelial migration of neutrophils [37]. All in all, this demonstrates that under inflammatory conditions, increased Trio expression and subsequent upregulation of adhesion molecules that mediate leukocyte extravasation [66].

## 7. The Potential Functions of Trio during Inflammation

As previously discussed, stimulation of HUVECs with the inflammatory stimuli LPS, TNFα and IFNγ upregulates Trio expression. The function of Trio in the endothelium indicates an important role in the regulation of the vascular permeability response during inflammation. Additionally, its downstream target, RhoA, not only has well established functions in multiple inflammatory processes, such as increased endothelial permeability but also in transendothelial migration [67,68,69]. Nevertheless, the link between Trio and the inflammatory response has not gained much attention. Trio was determined to be upregulated in multiple types of cancer, mainly implicated in relation to migration by increasing motility and, therefore, metastasis, but its function in inflammation is mostly unexplored [70,71,72]. One inflammatory disease that has been linked to Trio expression is Rheumatoid Arthritis (RA). Analysis of synovial biopsies of inflamed joints of RA patients displayed high expression of Trio, specifically in the endothelium, compared to patients with mild synovitis [33]. Furthermore, Trio colocalized with the adhesion molecules VCAM-1 and ICAM-1 in RA patient vessels, as was observed with immunofluorescence imaging [55]. This suggests Trio as a mediator of RA pathogenesis via the upregulation of adhesive molecules and, consequently, increasing leukocyte trafficking into the tissue.

In many chronic inflammatory diseases, thrombin generation increased [73,74,75]. Trio was suggested to be activated upon thrombin-induced Gα_q/11_ signaling [42]. This signaling pathway triggered RhoA activation and subsequently Rock/PRK activation, ultimately resulting in barrier destabilization and an increase in permeability through cytoskeletal remodeling and focal adhesion disruption [76]. However, if this pathway indeed signals via Trio, has yet to be determined. A disruption in flow that generates low shear stress is also known to induce inflammatory pathways, which is an essential feature in the development of atherosclerosis (a chronic inflammatory disease) [77]. As mentioned before, laminar shear stress induces alignment of endothelial cells by inducing Notch1 signaling, which requires localization of Trio at the cell membrane [49]. Under laminar flow, Trio could then be implicated to aid in the formation of a proper endothelial barrier and activation of Notch1. Furthermore, Notch signaling has shown anti-atherogenic properties by protecting endothelial disfunction against inflammatory stimuli [78]. To conclude, Trio aids in various inflammation-related processes, which possibly assist in the pathogenesis of chronic inflammatory diseases.

## 8. Clinical Implications of Trio Mutants

Abnormal RhoGEF function, in general, is linked to various human pathologies, including cancer [79,80,81,82]. While Trio is not linked to vascular diseases, it is most well known for its involvement in neuronal migration and axon outgrowth and guidance [83,84]. Trio mutations in the brain can result in multiple neuronal disorders ranging from schizophrenia to autism spectrum disorders and intellectual disorders. This is mostly due to mutations in the GEF1 domain or spectrin repeats of Trio, which either hamper or enhance activation of Rac1 [85,86,87,88]. Considering the function of Trio in migration, it is not surprising that mutations of this protein are found in multiple types of cancer [72,89]. For instance, Trio has been shown to be dysregulated in neuroblastoma; the authors implicated a decrease in Rac1 activation and increase in RhoA activation as an explanation for neurotigenesis [90]. In addition, one of the isoforms of Trio that is derived from alternative splicing of the Trio locus, called Trio-related transforming gene in ATL tumor cells (Tgat), is an oncoprotein that is found in T cell leukemias. This protein consists of only the GEF2 domain of Trio, but it contains a unique 15 amino acid long C-terminal end instead of the PH domain (Figure 1). These 15 amino acids are crucial for the transforming potential of Tgat. Furthermore, the expression of Tgat increases invasiveness in NIH3T3 fibroblasts, which was expected a result of RhoA activity. However, while inhibition of RhoGEF activity did reduce invasiveness, it was not completely abrogated. This indicates that the unique C-terminus also has invasive properties [91]. It was later found that Tgat activated the pro-inflammatory pathway NF-κB, by binding to the IκB complex. The C-terminal region of Tgat was crucial for binding to the complex. Nevertheless, GEF activity of Tgat was also necessary for NF-κB activation [92].

Contrary to Trio functioning abnormally, wildtype Trio has also shown to increase migration and, therefore, the metastasis of cancer. As was shown in colorectal cancer, Trio-mediated Notch activation induces RhoA activation and, therefore, enhanced cell migration. Interestingly, Rac1 activation was not affected, suggesting a specific role for RhoA in Trio-mediated invasiveness of these cancers [71]. In contrast, Trio was found to drive invadopodia disassembly via the Rac1-PAK1 axis, hinting towards a role for Rac1 in metestasis. This invadopodia disassembly was found to be necessary for correct migration and, therefore, metastasis because the cells showed impaired migration in 3D when Trio was removed [93].

It is not only the mutation of Trio itself but also a binding partner that can promote cancer. A common mutation of uveal melanoma is in the GNAQ genes, which results in the gain of function of G protein as subunits, such as Gα_q/11_ [94]. Feng et al. found that the activation of Rac1 and RhoA through this Gα_q/11_-Trio signaling pathway consequently activates the tumor promotor YAP. The regulation of the actin cytoskeleton played an important role for this activation as the accumulation of F-actin contributes to the nuclear translocation of YAP. YAP activation is frequently observed in multiple types of cancer, which is mostly believed to signal through the Hippo pathway [95]. However, this research indicates that this activation may also signal through Trio-Rac1/RhoA [70].

All in all, research that concentrates on Trio mutants highlights the importance of Trio-mediated Rac1 and RhoA activation in cell migration. This supports the previously indicated role of Trio in sprouting angiogenesis and the need for further research into this topic in the future.

## 9. Trio Inhibitors for Disease Prevention

The oncogenic potential of RhoGEFs spiked the interest for studying inhibitors as anti-cancer therapeutics [96,97]. While GTPases itself are widely distributed in tissues and are involved in multiple signaling cascades, RhoGEFs are usually activated downstream of a specific receptor and are less broadly expressed [98]. For Trio, several inhibitors have been identified over the years, both for the GEF1 and GEF2. The peptide TRIPα was the first specific RhoGEF inhibitor discovered. It inhibits RhoA via the GEF2 domain of Trio and has no influence on RhoG activation [99]. Shortly after, Blangy and coworkers preformed a screen in yeast to find potential Trio GEF1 inhibitors [100]. They found one and two chemical analogues that inhibited RhoG activation in vitro, however, it turned out to be toxic in vivo. When returning to the screen, they found ITX3, a specific Trio inhibitor that interferes with Rac1/RhoG activation and is non-toxic in vivo [101]. This inhibitor is very promising because it neither interferes with GEF2-mediated RhoA activation nor inhibits other GEFs such as Tiam1 and Vav2. It specifically inhibits GEF1, but it leaves the availability of RhoG and Rac1 intact, allowing them to be activated by different GEFs [101]. Whether these inhibitors can be used to treat cancer remains to be investigated, but it is a promising tool.

## 10. Conclusions

Over the years, it has become evident that Trio functions as a molecular regulator of multiple endothelial functions. This is tightly regulated by multiple mechanisms, such as auto-inhibition, inhibitory proteins and varying interactions, to ensure proper barrier function and, thus, vascular integrity. Dysregulation of Trio signaling may possibly be involved in the development or severity of chronic inflammatory diseases. However, the wide-ranging functions of Trio in the endothelium from remodeling of the actin cytoskeleton to regulating the expression of inflammatory adhesion molecules render Trio a challenging RhoGEF to study. We speculate that, depending on the environment and the need of the endothelium at that specific moment in time, a different activation of Trio is required in order to exert one of its many functions. For example, Trio may be activated during inflammation, thereby regulating endothelial permeability and limiting vascular damage.

Many questions remain to be elucidated and more insight into the versatile role of Trio in the endothelium will provide greater understanding of pathologic conditions in which the endothelial barrier is compromised.

## Figures and Tables

**Figure 1 ijms-22-10168-f001:**
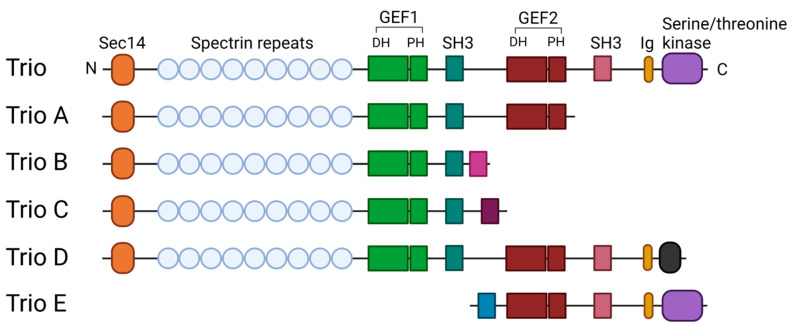
Schematic representation of Trio isoforms. The full-length Trio contains a Sec14 domain, several Spectrin repeats, two GEF domains, two S SRC Homology 3 Domains (SH3) and a serine/threonine kinase domain. Other isoforms arise due to alternative splicing and contain a mix of these domains. The pink, purple and turquoise box represent specific amino acids unique to the isoform. The figure was created with BioRender.

**Figure 2 ijms-22-10168-f002:**
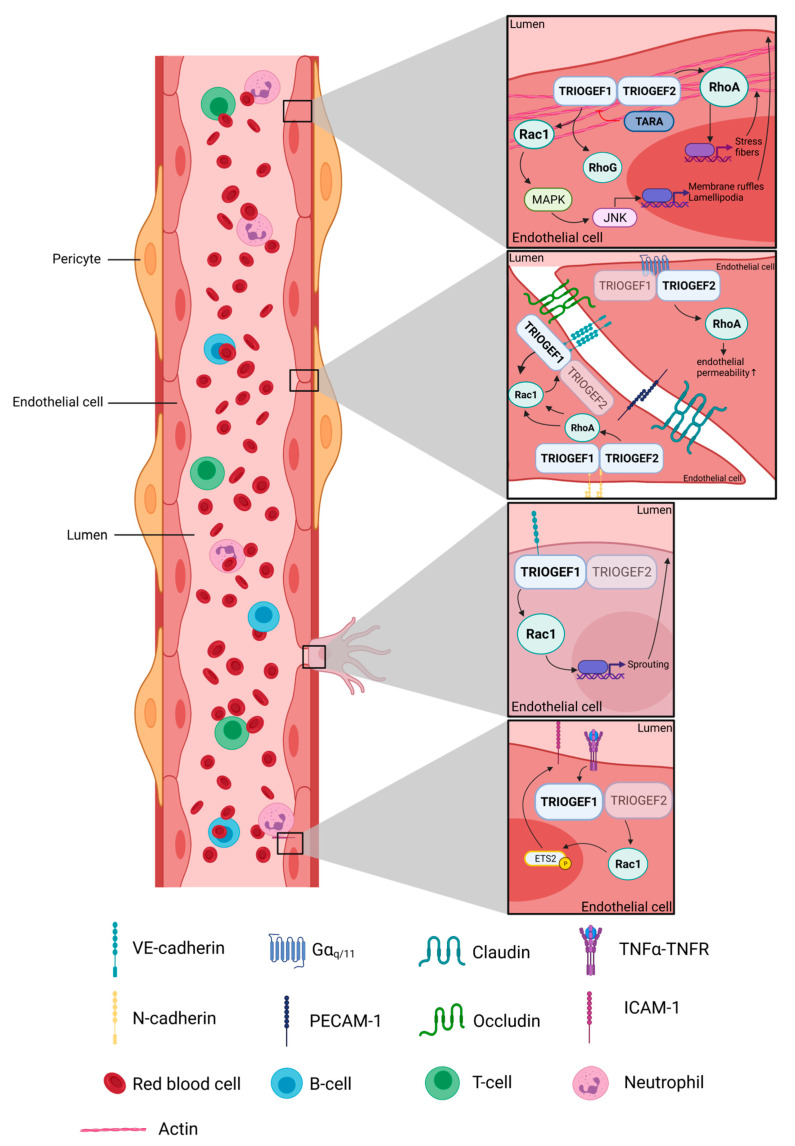
Functions of Trio in the endothelium. The two GEF domains of Trio can activate either Rac1/RhoG or RhoA. This activation can induce a multitude of functions, of which actin cytoskeleton remodeling, endothelial barrier regulation, sprouting and adhesion molecule expression enabling leukocyte transendothelial migration are shown here. If TrioGEF is depicted in a faint color, this means that the domain does not play a role in this specific pathway. The figure was created with BioRender.

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
