# Peer review of "The RhoGEF Trio: A Protein with a Wide Range of Functions in the Vascular Endothelium"

_ijms, 2021, doi:10.3390/ijms221810168_

Round 1

Reviewer 1 Report

The manuscript submitted by Kempers et al., covers several aspects of Trio's biology but mainly its role as a guanine exchange factor (GEF). The review focuses on the functions of Trio in the endothelium but also extrapolates data from other cell types to better understand the role Trio may play in vascular biology. It gathers data from the literature in a reliable and therefore useful way.

A number of changes could improve the text and figures, as suggested below:

- Overall, the review is mainly focused on the GEF function of Trio. There is little mention on the involvement of other domains (such as for example spectrin repeats, SH3 domains, the Ser/Thr kinase domain) in Trio biology. If this is the choice of the authors, some references could be provided to guide readers in their exploration of other functions of Trio. This is particularly relevant for the serine/threonine kinase activity depicted in figure 1 but for which there is only one sentence, just mentioning the existence of this enzymatic activity, in the main text.

- many of the references are reviews when some could be original articles. For instance: page 1 and 2, lines 45-47: “The third enzymatic domain of Trio is a serine/threonine kinase domain, which specifically phosphorylates serine/threonine amino acid residues [5].”

- More details should be given in the legend of figure 1, for instance: explain SH3, explain the colored rectangles which are only found in trioB and trioE…

- Page 2, lines 51-52: “Our preliminary data show that at least TrioB is expressed in endothelial cells”.

Please specify that these are unpublished data and also which method was used (qRT-PCR, WB...).

- Page 3, lines 88-91: “As was shown more than 20 years ago, the enzymatic activity of the N-terminal TrioGEF1 induces Jun Kinase (JNK) activation through the mitogen activated protein kinase (MAPK) pathway, both also recognized as downstream targets of Rac1 [7]. “

Unclear sentence

- Page 3, lines 91-92: “This results in the formation of membrane ruffles and lamellipodia, a key characteristic of migrating cells (Figure 2) [26].”

There is no obvious connection between this sentence and figure 2. There is no actin / cytoskeletal structure shown in figure 2.

- Page 3, lines 95-97: “RhoG has a higher binding affinity for Trio than Rac1, potentially in more effective GDP/GTP switching [28][29].”

It seems that the verb is missing.

- Page 3, line 109-110: “Besides RhoG and Rac1, RhoA can be activated by the GEF2 domain of Trio. This activation contributes to the formation of actin stress fibers (Figure 2) [26].”

There is no obvious connection between this sentence and figure 2. There is no actin / cytoskeletal structure shown in figure 2.

- Figure 2: The lettering in the right-hand panels in figure 2 is difficult to read. The size of the letters can be increased by enlarging these panels.

First panel on the top right is awkward. It seems that stress fibers and membrane ruffles lamellipodia are nuclear events. Same comment applies for the third left panel on the right, indicating sprouting on top of what seems to be a nucleus.

Trio is shown too small is these panels focusing on Trio regulatory pathways and associated cellular events.

- Page 3, line 131-132. “Enhanced metabolic requirements of tissues can lead to enlargement of endothelial cells, causing a larger diameter of the vessel.”

This sentence is not clear if one considers the following ones. Do the authors mean “Enhanced tissue metabolic requirements result from endothelial cell hypertrophy associated with vessel diameter enlargement.”?

Please, clarify the meaning in an unambiguous way.

- Page 4, lines 140-142: “In Hela cells, Trio was found to inhibit cytokinesis, the last stage of cell proliferation [36]. This occurred by activation of Rac1, which negatively regulates cytokinesis [36].”

Cytokinesis is related to mitosis (not proliferation). How is the effect of Trio on cytokinesis related to increase in cell size? Is there a cytokinesis blockade during vessel enlargement?

In addition, ref 36 does not report that Trio inhibits cytokinesis. Ref 36 shows that Trio functions as a GEF of Rac1 during cell division. Rac1 negatively regulates the assembly and constriction of the contractile ring during cytokinesis. Rac1 activation needs to be inhibited at the cleavage furrow and this is normally mediated by MgcRacGAP. Trio depletion rescues the cytokinesis failure induced by MgcRacGAP depletion in HeLa cells.

Please rephrase these sentences.

- Page 6, lines 170-171. “Another study showed that the activation of N-cadherin in endothelial cells induces the formation of a protein complex with Trio at the cell surface”.

Do the authors mean at the plasma membrane?

The pericytes that are mentioned on page 6 (lines 175-179) could be pointed out on the schema representing a vessel in figure 2.

- Page 7, line 201.  The authors refer to the C-terminal end of Trio as TrioC. This is confusing as Trio C is also the name given to one of Trio splice variant (figure 1).

- Page 7, section “migration and sprouting angiogenesis require the GEF1 domain of Trio”, line 241-251. The content of these 10 lines is speculative and this portion of the text could be reduced.

- Page 8, lines 283-286. “All in all, this demonstrates that under inflammatory conditions, increased Trio expression and subsequent activation mediates leukocyte extravasation through the induction of membrane structures that have been believed to be docking structures [61].”

Please revise this sentence and give more details (including reference (s) for increase Trio expression under inflammatory conditions.

- Page 8, lines 301-302: “This indicates Trio as a mediator of RA pathogenesis, by increasing leukocyte trafficking into the tissue.”

Please mitigate this statement.

- Page 9, lines 341-350.

This section is unclear and should be re-written.

Author Response

We thank the reviewer for his/her thoughtful comments to our manuscript. We have addressed all comments to the best of our ability and listed them in a point-to-point manner below. Please find the revised manuscript in the attachement. 

Reviewer 1

The manuscript submitted by Kempers et al., covers several aspects of Trio's biology but mainly its role as a guanine exchange factor (GEF). The review focuses on the functions of Trio in the endothelium but also extrapolates data from other cell types to better understand the role Trio may play in vascular biology. It gathers data from the literature in a reliable and therefore useful way.

 A number of changes could improve the text and figures, as suggested below:

 - Overall, the review is mainly focused on the GEF function of Trio. There is little mention on the involvement of other domains (such as for example spectrin repeats, SH3 domains, the Ser/Thr kinase domain) in Trio biology. If this is the choice of the authors, some references could be provided to guide readers in their exploration of other functions of Trio. This is particularly relevant for the serine/threonine kinase activity depicted in figure 1 but for which there is only one sentence, just mentioning the existence of this enzymatic activity, in the main text.

We added the original references to the manuscript for the studies related to Trio. In addition, we added a small section about the different domains of Trio.

- many of the references are reviews when some could be original articles. For instance: page 1 and 2, lines 45-47: “The third enzymatic domain of Trio is a serine/threonine kinase domain, which specifically phosphorylates serine/threonine amino acid residues [5].”

We changed the references to original articles when related to Trio.

 - More details should be given in the legend of figure 1, for instance: explain SH3, explain the colored rectangles which are only found in trioB and trioE…

We expanded the figure legend to contain all the information about the different domains depicted in the Figure. 

 - Page 2, lines 51-52: “Our preliminary data show that at least TrioB is expressed in endothelial cells”.

Please specify that these are unpublished data and also which method was used (qRT-PCR, WB...).

We added between brackets that the data was unpublished and added the method by which this was investigated (PCR).

- Page 3, lines 88-91: “As was shown more than 20 years ago, the enzymatic activity of the N-terminal TrioGEF1 induces Jun Kinase (JNK) activation through the mitogen activated protein kinase (MAPK) pathway, both also recognized as downstream targets of Rac1 [7]. “

Unclear sentence

We rewrote the sentence to make it easier to read.

- Page 3, lines 91-92: “This results in the formation of membrane ruffles and lamellipodia, a key characteristic of migrating cells (Figure 2) [26].”

There is no obvious connection between this sentence and figure 2. There is no actin / cytoskeletal structure shown in figure 2.

Figure 2 was adapted. Actin structures were added to the figure and more clarification was given about what is depicted.

 - Page 3, lines 95-97: “RhoG has a higher binding affinity for Trio than Rac1, potentially in more effective GDP/GTP switching [28][29].”

It seems that the verb is missing.

The sentence was adjusted.

 - Page 3, line 109-110: “Besides RhoG and Rac1, RhoA can be activated by the GEF2 domain of Trio. This activation contributes to the formation of actin stress fibers (Figure 2) [26].”

There is no obvious connection between this sentence and figure 2. There is no actin / cytoskeletal structure shown in figure 2.

Stress fibers were added to the upper right panel.

 - Figure 2: The lettering in the right-hand panels in figure 2 is difficult to read. The size of the letters can be increased by enlarging these panels.

First panel on the top right is awkward. It seems that stress fibers and membrane ruffles lamellipodia are nuclear events. Same comment applies for the third left panel on the right, indicating sprouting on top of what seems to be a nucleus.

Trio is shown too small is these panels focusing on Trio regulatory pathways and associated cellular events.

We agree with the reviewer that indeed, the text was too small. Figure 2 was edited to contain stress fibers and the ‘event’ stress fibers now links to actual stress fibers. The panels on the right were increased to make it easier to see.

In addition, we added more information about the cell and events displayed to the figure.

- Page 3, line 131-132. “Enhanced metabolic requirements of tissues can lead to enlargement of endothelial cells, causing a larger diameter of the vessel.”

This sentence is not clear if one considers the following ones. Do the authors mean “Enhanced tissue metabolic requirements result from endothelial cell hypertrophy associated with vessel diameter enlargement.”?

Please, clarify the meaning in an unambiguous way.

We apologize for the inconvenience. In fact, we meant it the other way around. Please allow us to explain: If organs are in distress or growing the demand for blood will increase. To respond to this, the vessels enlarge, leading to an increased diameter and more flow in the organs. Interestingly, the endothelial cells do not divide/proliferate to give rise to these larger vessels, but the cells themselves become larger/hypertrophied.

To avoid any confusion, we changed to sentence to make this clearer.

 - Page 4, lines 140-142: “In Hela cells, Trio was found to inhibit cytokinesis, the last stage of cell proliferation [36]. This occurred by activation of Rac1, which negatively regulates cytokinesis [36].”

Cytokinesis is related to mitosis (not proliferation). How is the effect of Trio on cytokinesis related to increase in cell size? Is there a cytokinesis blockade during vessel enlargement?

In addition, ref 36 does not report that Trio inhibits cytokinesis. Ref 36 shows that Trio functions as a GEF of Rac1 during cell division. Rac1 negatively regulates the assembly and constriction of the contractile ring during cytokinesis. Rac1 activation needs to be inhibited at the cleavage furrow and this is normally mediated by MgcRacGAP. Trio depletion rescues the cytokinesis failure induced by MgcRacGAP depletion in HeLa cells.

Please rephrase these sentences.

We deleted this sentence with the references, as it does not add to the clarity of the Ms.

 - Page 6, lines 170-171. “Another study showed that the activation of N-cadherin in endothelial cells induces the formation of a protein complex with Trio at the cell surface”.

Do the authors mean at the plasma membrane?

Indeed, we do. We accordingly changed this in the manuscript.

The pericytes that are mentioned on page 6 (lines 175-179) could be pointed out on the schema representing a vessel in figure 2.

This was added in the text.

- Page 7, line 201.  The authors refer to the C-terminal end of Trio as TrioC. This is confusing as Trio C is also the name given to one of Trio splice variant (figure 1).

We agree. Therefore, we replaced TrioC to  “the C-terminal end of Trio”.

 - Page 7, section “migration and sprouting angiogenesis require the GEF1 domain of Trio”, line 241-251. The content of these 10 lines is speculative and this portion of the text could be reduced.

We reduced this section.

 - Page 8, lines 283-286. “All in all, this demonstrates that under inflammatory conditions, increased Trio expression and subsequent activation mediates leukocyte extravasation through the induction of membrane structures that have been believed to be docking structures [61].”

Please revise this sentence and give more details (including reference (s) for increase Trio expression under inflammatory conditions.

We added a sentence with reference stating that Trio expression is increased by incubation with TNFα. We reduced the final sentence of this paragraph.

 - Page 8, lines 301-302: “This indicates Trio as a mediator of RA pathogenesis, by increasing leukocyte trafficking into the tissue.”

Please mitigate this statement.

Done

 - Page 9, lines 341-350.

This section is unclear and should be re-written.

Done

Reviewer 2 Report

In general, the authors should think about re-arrange the sections, for example, from broad area to narrow area. It seems the manuscript is not well organized, and the contents are spread out in the manuscript. 

In lines 53-54 (reference 8), the authors ran scRNAseq in ECs from many different tissues. Is there any difference in expression levels among the tissue in the reference? For example, Trio would be more express in brain vs. soleus. It is well known that brain endothelial cell has more permeability related phenotypes than other endothelial cells. If you found the results, it would be good to address Trio expression levels among the tissues.

In lines 75-76, the authors should describe more about the endothelial cells? What endothelial cells? Human, mouse, or bovine / artery or vein / brain, muscle, or cardiac / etc..  
Also, it would be good to know what other cell types people have studied for Trio, and provide a brief description of what they have found.

In lines 105-16, to conclude that activation of RhoG or Rac1 depends on the environments and specific cell types, the authors need to provide more information about environments and cell types. 

In lines 140-142 (reference 36), the author should take out this reference or provide a better connection between vascular cells and hela cells because this is not about vascular cell…and there is no link to the vascular cells. 

In figure 2, I don’t know it happens only on my screen, but figure 2 looks blurry. It is hard to see the texts in the images. Also, the authors put the label of the image on the bottom of the figures like VE-Cad, N-cad, etc…however, the authors should provide more of this for example, in the left panel (long vessel), it seems hemoglobin, macrophage, and neutrophils, etc., but it is not clear because there are no labels. On the right panels (enlarged images), please clarify where lumen, endothelial cell, smooth muscle cell, etc. The author used faint texts for TRIOGEF2 or 1 in the figures….it is really hard to see and not clear what is this mean….

In line 149, the title of this section is part of the vascular cell phenotype. That is why I asked to think about re-organizing like non-vascular cell, vascular cell, vascular endothelial cell-related disease, etc. Also, since brain endothelial cell phenotype is highlight related to barrier function, it would be good to have a section for the role of the Trio in brain endothelial cells.

In line 180, is there any literature about pathophysiological conditions like oscillatory or disturbed flow?  If there is, please add. 

In the disease sections, the author talks about Trio as a potential therapeutic target for cancer. This is a discrepancy between the contents that the authors have built up (role of Trio in the vascular endothelial cell) and the disease model. Generally, endothelial cell-related treatment can be a therapeutic target for cancer to regulate angiogenesis, but there is nothing about angiogenesis or endothelial cell-specific things. Therefore, it would be better to provide the role of Trio in disease models like atherosclerosis, stroke, etc., instead of cancer. 

Author Response

We thank the reviewer for his/her thoughtful comments to our manuscript. We have addressed all comments to the best of our ability and listed them in a point-to-point manner below. The reviewer raised some important questions. However, some of the questions raised are not yet investigated in the lab. We hope that with this review we implicate the importance of Trio in the vasculature and that we will be able to answer these questions in the future.  

Reviewer 2

In general, the authors should think about re-arrange the sections, for example, from broad area to narrow area. It seems the manuscript is not well organized, and the contents are spread out in the manuscript. 

We understand the point of the reviewer. We moved around some sections, to make it easier to read. The difficulty remains that Trio function in endothelial cells is to this date not very extensively researched, so a lot remains unknown.

To hypothesize or show why we suspect Trio to be important in endothelial cells we have included studies that were done in other cell types. We understand the point raised by the reviewer. Therefore, we have re-organized the Ms by a section covering the vascular function and the more general role Trio may play in cell biology. 

In lines 53-54 (reference 8), the authors ran scRNAseq in ECs from many different tissues. Is there any difference in expression levels among the tissue in the reference? For example, Trio would be more express in brain vs. soleus. It is well known that brain endothelial cell has more permeability related phenotypes than other endothelial cells. If you found the results, it would be good to address Trio expression levels among the tissues.

When using the online tool to see the expression levels of Trio in endothelial cells from different organs, we found little differences. There seems to be a bit more Trio expression in the testis, but the expression is still low. I would like to show a picture but the box doesn't allow that. 

In lines 75-76, the authors should describe more about the endothelial cells? What endothelial cells? Human, mouse, or bovine / artery or vein / brain, muscle, or cardiac / etc..  
Also, it would be good to know what other cell types people have studied for Trio, and provide a brief description of what they have found.

We thank the reviewer for these valuable ideas. We have added extra information to the review: what type of endothelial cells the studies we refer to were performed. As our focus was on Trio in the vascular endothelium, we didn’t want to describe to much about Trio in different cell types. Other reviews are more thorough on this and we have added these for the reader.

In lines 105-16, to conclude that activation of RhoG or Rac1 depends on the environments and specific cell types, the authors need to provide more information about environments and cell types. 

We have added more information on extracellular matrix proteins, like fibronectin, in this paragraph.

In lines 140-142 (reference 36), the author should take out this reference or provide a better connection between vascular cells and hela cells because this is not about vascular cell…and there is no link to the vascular cells. 

We apologize for this. The reference is deleted.

In figure 2, I don’t know it happens only on my screen, but figure 2 looks blurry. It is hard to see the texts in the images. Also, the authors put the label of the image on the bottom of the figures like VE-Cad, N-cad, etc…however, the authors should provide more of this for example, in the left panel (long vessel), it seems hemoglobin, macrophage, and neutrophils, etc., but it is not clear because there are no labels. On the right panels (enlarged images), please clarify where lumen, endothelial cell, smooth muscle cell, etc. The author used faint texts for TRIOGEF2 or 1 in the figures….it is really hard to see and not clear what is this mean….

The blurriness may be due to the resolution of the PDF merge. We ha ve increased the pixels for the figure to increase its resolution and make it less blurry. 

We also added more information about the cells that are depicted in the legend. When the GEF2 domain is faint, this means it does not play a dominant role in this vascular event. We have clarified this now in the figure legend.

In line 149, the title of this section is part of the vascular cell phenotype. That is why I asked to think about re-organizing like non-vascular cell, vascular cell, vascular endothelial cell-related disease, etc. Also, since brain endothelial cell phenotype is highlight related to barrier function, it would be good to have a section for the role of the Trio in brain endothelial cells.

We agree with the reviewer that Trio could potentially be important in the tight barrier of the brain. We do show that Trio is important in regulation of the barrier function in vitro, but these studies are mostly done in HUVEC. How these results translate to the brain are not known. To our knowledge Trio has not been investigated in the brain with the focus on endothelial cells. As Trio is well known to be involved in neuronal growth, most research regarding the brain has focused on that part. We added a small section about the possible role of Trio in the brain in the paragraph about barrier regulation.

In line 180, is there any literature about pathophysiological conditions like oscillatory or disturbed flow?  If there is, please add. 

To the best of our knowledge, there is no literature out there about disturbed flow and Trio function. We would hypothesize that disturbed flow would influence the Trio function/activity, as it is known that Trio associates with VE-cadherin and the LAR domain. How this disturbed Trio function would manifest is unknown.

In the disease sections, the author talks about Trio as a potential therapeutic target for cancer. This is a discrepancy between the contents that the authors have built up (role of Trio in the vascular endothelial cell) and the disease model. Generally, endothelial cell-related treatment can be a therapeutic target for cancer to regulate angiogenesis, but there is nothing about angiogenesis or endothelial cell-specific things. Therefore, it would be better to provide the role of Trio in disease models like atherosclerosis, stroke, etc., instead of cancer. 

We could not find any information about the role of Trio in vascular diseases. We think Trio could play an important role, but there is no literature to support this.

Therefore, we added the section about Trio in cancer and other diseases, to highlight that Trio is important and needs to function normally. With this review, we wish to spike the interest of endothelial biologist to consider Trio as an important player in the vasculature.

Round 2

Reviewer 2 Report

Thank you for all the responses to my comments. 
Overall, the manuscript looks good, but I have some minor comments. 

In line 58, VE-cadherin (and endothelial marker)...do you mean an endothelial marker? If it is, please correct it. Also, an endothelial cell marker would be more appropriate. 

In line 306, please check the format of reference.

In line 359, "not yet linked to vascular diseases..." looks weird because no reference shows the relationship between Trio and vascular disease. If you have it, please put it in here. But if you don't have it, the sentence should be amended. Take out "yet"...

In line 380, "dysfunctioning Trio"...dysfunctioning is not the usual term. Would you please amend this sentence as well?

Author Response

Than you for reading our manuscript carefully! We changed the comments and mistakes!

Thank you for all the responses to my comments. 
Overall, the manuscript looks good, but I have some minor comments. 

In line 58, VE-cadherin (and endothelial marker)...do you mean an endothelial marker? If it is, please correct it. Also, an endothelial cell marker would be more appropriate. 

We indeed mean an endothelial cell marker, this was changed in the manuscript.

In line 306, please check the format of reference.

Done! Thank you for spotting our mistake!

In line 359, "not yet linked to vascular diseases..." looks weird because no reference shows the relationship between Trio and vascular disease. If you have it, please put it in here. But if you don't have it, the sentence should be amended. Take out "yet"...

This was taken out.

In line 380, "dysfunctioning Trio"...dysfunctioning is not the usual term. Would you please amend this sentence as well?

This was changed to:  “On the contrary to Trio functioning abnormally, wildtype Trio” 
